# The Role of Paraclinical Investigations in Detecting Inflammation in Children and Adolescents with Obesity and Metabolic Syndrome

**DOI:** 10.3390/life14091206

**Published:** 2024-09-23

**Authors:** Mihaela-Andreea Podeanu, Ștefănița Bianca Vintilescu, Claudiu Marinel Ionele, Raluca Elena Sandu, Carmen Elena Niculescu, Mirela-Marinela Florescu, Mioara Desdemona Stepan

**Affiliations:** 1Doctoral School, University of Medicine and Pharmacy of Craiova, 200349 Craiova, Romania; podeanu.andreea11@gmail.com; 2Department of Infant Care, Pediatrics and Neonatology, University of Medicine and Pharmacy of Craiova, 200349 Craiova, Romaniadesdemona.stepan@umfcv.ro (M.D.S.); 3Department of Gastroenterology, University of Medicine and Pharmacy of Craiova, 200349 Craiova, Romania; 4Department of Biochemistry, University of Medicine and Pharmacy of Craiova, 200349 Craiova, Romania; ralucasandu80@gmail.com; 5Department of Pathology, University of Medicine and Pharmacy of Craiova, 200349 Craiova, Romania; mirela.florescu@umfcv.ro

**Keywords:** obesity, metabolic syndrome, inflammation, inflammatory indexes, cardioembolic indexes, children, adolescents

## Abstract

Obesity is linked to the increasing prevalence of metabolic syndrome (MetS), even among the pediatric population. Some inflammatory and cardioembolic indexes derived from routine laboratory tests have captivated the attention of the medical community. Objectives: The aim of our study was to evaluate whether these markers are effective in distinguishing varying degrees of obesity and MetS in children and adolescents. Methods: We conducted a retrospective study. A total of 71 children and adolescents, aged between 6 and 16, were included in the study. Among them, 5 were overweight, 35 had obesity, and 31 had severe obesity. According to the NCEP ATP III criteria, 32 individuals had Metabolic Syndrome (MetS), while 39 did not have MetS. Results: The MetS positive group had higher values of TG/HDL-C (*p* < 0.001), TC/HDL-C (*p* < 0.001), MHR (*p* = 0.015), LHR (*p* = 0.001), NHR (*p* = 0.001), atherogenic index of plasma (*p* < 0.001), and PHR (*p* < 0.001). ESR, NLR, PLR, and SII did not progressively increase with the number of MetS criteria. The ROC curve analysis demonstrated that markers such as TG/HDL-C, the atherogenic index of plasma, TC/HDL-C, LHR, NHR, and PHR were effective in identifying MetS in children and adolescents with obesity. Conclusions: In conclusion, we determined that some novel inflammatory and cardioembolic indexes are useful in assessing MetS and obesity in children and adolescents.

## 1. Introduction

For many decades, the topic of weight dysregulation in the pediatric population has had its atten tion set on the spectrum of malnutrition [1]. Many programs have been focusing on ending it, especially for children in low-income countries [2]. Currently, with urbanization [3] and the increase in the prevalence of processed food and its availability all over the world [4], a new enemy is rapidly emerging in the form of overweight and obesity, with a devastating impact on the general health of both adult and pediatric populations [5]. In both middle-income and low-income countries, the percentage of children and adolescents with excess weight builds up every year according to statistics [6], with a coexistence of malnutrition in the same populations, a phenomenon called “the double burden of nutrition” [2,7].

According to the Centers for Disease Control and Prevention (CDC), before the COVID-19 pandemic, the prevalence of obesity among the pediatric population was on an ascending slope [8]. Epidemiologic reports of obesity prevalence in various countries are showing a rapid increase in weight gain across all age groups and socioeconomic categories [9,10,11,12]. As some studies report, children that are affected by obesity tend to become adults not only with obesity, but with morbid obesity, which considerably raises predisposition to chronic medical issues over the years [13]. Also, exposure to obesogenic factors during early childhood predisposes one to an increased risk for metabolic syndrome (MetS) in adults [14].

Obesity is mainly characterized by excessive fat accumulation, with a complex ethology, involving genetic and environmental factors [15]. Its link with MetS, a pathology defined by central obesity, hypertension, insulin resistance, and dyslipidemia, is well known [16]. Even though there is a direct association, not all children with obesity develop MetS, with the literature describing these persons as metabolically healthy obese [17].

Even though the fact that MetS is affecting more and more children and adolescents, is undeniable [18,19], the criteria used to diagnose this pathology in the pediatric population is still controversial because it is not clear yet which definition is better used, which are the right cutoff points for each criterion, what is the clinical significance, and how accurate it is in determining the risk of developing cardiovascular diseases (CVD) and type 2 diabetes mellitus (T2DM) [20].

Both simple obesity and MetS have been associated with a degree of low-grade inflammation [21,22]. The clear physiopathology behind MetS is not yet fully understood, but more and more reports show an increase in proinflammatory cytokines [23,24] due to the fact that metabolic disturbances lead to immune activation especially in the adipose tissue [25,26,27,28] and individuals often present with elevated plasma markers of chronic low-grade inflammation [29,30], which can lead to a lot of serious consequences in the future [31,32,33,34,35,36].

Studies indicate that the atherosclerotic process can start during childhood, with atherogenic dislipidemia becoming more prevalent in the younger population affected by obesity. Also, the presence of atherogenic dislipidemia, along with MetS, significantly elevates the risk of developing CVD sooner or later [37].

In the adult population, good predictors of cardiovascular risk assessment are lipid ratios, such as the triglyceride (TG) to high-density lipoprotein cholesterol (HDL-C) ratio (TG/HDL-C) and total cholesterol (TC) to HDL-C ratio (TC/HDL-C) [38]. Also, the TG/HDL-C ratio was used to identify individuals with insulin resistance [39] and to detect carotid artery intima–media thickness in both adults and children [40,41]. The non-HDL-C showed a strong association with MetS risk for CVD [42,43].

Many studies have investigated the best inflammatory markers to detect the low-grade inflammation associated with obesity and MetS. However, in clinical practice, these markers must be widely available, reproducible, and, very importantly, affordable for routine use in most healthcare settings. When a child or adolescent with overweight or obesity is examined, basic paraclinical tests also include a complete blood count (CBC), TG, TC, and HDL-C, as they are mandatory for sustaining the diagnosis. Therefore, novel inflammatory and cardioembolic markers such as the neutrophils to lymphocytes ratio (NLR), platelets to lymphocytes ratio (PLR), neutrophils to HDL-C ratio (NHR), monocytes to HDL-C ratios (MHR), lymphocyte to HDL-C ratio (LHR), platelets to HDL-C ratio (PHR), Systemic Immune-Inflammation Index (SII), and the Atherogenic index of plasma are interesting tools used by clinicians [44].

The aim of our study was to investigate if these markers are reliable in differentiating between children and adolescents with obesity with different grades of obesity and MetS.

## 2. Materials and Methods

### 2.1. Study Design

We conducted a retrospective study, in which we assessed children and adolescents aged between 6 and 16 years, diagnosed with obesity or overweight. A systematic search was conducted in the archives of the Clinical Emergency County Hospital of Craiova, and patient data were collected from the medical records. The inclusion criterion for the initial search was represented by the presence of overweight or obesity, determined by a body mass index (BMI) ≥ 85th percentile, calculated in accordance with the CDC criteria in relation to age and gender. Our search was restricted to the last three and a half years (January 2021 to July 2024). The International Classification of Diseases, Tenth Revision (ICD-10) codes for the diagnosis of obesity (E66.0, E66.3, E66.8, and E66.9) and a manual search in the patient’s record were used to identify the subjects eligible for our study.

After that, we excluded the patients that were younger than 6 years of age and older than 16 years of age. Also, patients with T2DM, secondary obesity (genetic syndromes, monogenic obesity, endocrine pathology), chronic disease or history of acute infection in the last 3 months (including COVID-19), children with leukopenia (<4 × 10^9^/L), leukocytosis (>13 × 10^9^/L), anemia or thrombocytopenia (<150 × 10^9^/L), a history of drug use that could affect the CBC parameters or bodyweight were not taken into consideration. From the remaining patients, we only extracted the data from the patients that had all the parameters needed for our analysis.

The parameters of interest were age, gender, weight, height, abdominal perimeter or mention about the percentile of the abdominal perimeter, systolic blood pressure (SBP), diastolic blood pressure (DBP), fasting blood glucose (referred as glycemia), triglycerides, total cholesterol (TC), high-density lipoprotein cholesterol (HDL-C), erythrocyte sedimentation rate (ESR), hemoglobin, leucocytes (absolute value), lymphocytes (absolute value), neutrophils (absolute value), platelets (absolute value), basophiles (absolute value) and, where it was available, C-reactive protein (CRP), which was only used as a double check for acute disorders, and not used as a parameter for the current research.

The data were extracted by the investigators in a predefined Excel worksheet. No personal data of the patients was stored. For each individual, a random code was assigned instead of name in order to keep order, but it served no purpose in the further processing of data.

### 2.2. Subjects Included in the Study

The final lot of study was composed of 71 children and adolescents with overweight and obesity and a waist circumference greater than the 90th percentile for age and sex. In this group, there were 48 boys and 23 girls.

None of the children and adolescents included in the study were on any medication or diagnosed with any other acute or chronic condition, including COVID-19, at the moment of the examination according to the data stated in their medical file. All patients were Caucasians (Non-Hispanic Whites) and originated from the south-west of Romania (Oltenia region).

Subjects were divided into groups according to the presence or absence of MetS (MetS− and MetS+ group), the category of BMI according to age and sex using the CDC criteria (overweight, obesity, and severe obesity) and according to the number of MetS diagnosis criteria (1 MetS diagnosis criteria, 2 MetS diagnosis criteria, 3 MetS diagnosis criteria, 4 MetS diagnosis criteria, and 5 MetS diagnosis criteria).

### 2.3. Clinical Assessment and Anthropometric Measurements

The patient’s history was assessed in order to identify the eligible cases, and the epidemiological data represented by age and gender were recorded.

The weight and height were important anthropometric data that we collected, which was performed using standardized measurement methods. For height measurement, a thaliometer with a scale of 1 mm, and for weight, a dedicated monitor for determining weight (gradation of 100 g) was used.

BMI was calculated using Excel, for the minimization of human errors, using the formula: BMI = Weight/Height^2^. Weight was measured in kilograms (kg) and height was measured in meters (m).

The percentile for BMI was calculated on the official website of the CDC and required data regarding the measurement units (we used the metric units), the gender, age, height and weight of every subject [45]. The BMI and the corresponding percentile and BMI category were automatically displayed. Obesity and overweight were defined according to BMI as it follows:Overweight: 85th to 94th percentile;Obesity: 95th to 99th percentile or BMI > 30 kg/m^2^;Severe obesity: >120 percent of the 95th percentile or BMI > 35 kg/m^2^.

Blood pressure was measured with an electronic tensiometer after a period of 5 to 10 min of rest. In order to determine the percentile for blood pressure, we used an online platform widely used for this purpose [46,47]. This software uses the guidelines of the CDC and takes into account the gender, height, and systolic and diastolic blood pressure. The 90th percentile of blood pressure was considered the threshold between normal and high blood pressure. Even though a value between the 90th and 95th percentile is defined as prehypertension, between 95th and 99th as grade I hypertension, and over 99th as grade II hypertension, for the current study, and in concordance with the diagnostic NCEPT-ATP III criteria for MetS in children and adolescents [48], we considered all the values above the 90th percentile as hypertension.

### 2.4. Laboratory Analysis

A blood sample was obtained after a minimum of 8 h of fasting, in the morning, by phlebotomy by a qualified nurse. All the probes were processed by automatic, standardized laboratory techniques.

### 2.5. Definitions Used and Formulas

For the definition of MetS, we used the NCEP-ATP III adapted for children and adolescents. For a positive diagnosis, at least 3 out of 5 of the following criteria should be met:Central obesity with WC ≥ 90th percentile for age and sex,TG ≥ 110 mg/dL,HDL-C ≤ 40 mg/dL,Blood pressure (systolic or diastolic) ≥ 90th percentile for age and sex,Fasting blood glucose ≥ 100 mg/dL [48].

In our study, all the participants met the criteria for central obesity. For a positive MetS diagnosis, participants needed to meet at least two additional criteria, other than central obesity.

All the children and adolescents with BMI modifications included in the study were divided into 2 groups according to the presence and absence of MetS (MetS− and MetS+ groups). Additionally, to assess the cumulative impact of the severity MetS on the inflammatory and cardioembolic indexes, we divided the subjects into groups based on the number of MetS diagnostic criteria they met, ranging from 1 to 5.

The following formulas were used for the calculation of the inflammatory indexes:Neutrophil to lymphocyte ratio (NLR) = neutrophil count (10^9^/L)/lymphocyte count (10^9^/L);Platelets to lymphocytes ratio (PLR) = platelets count (10^9^/L)/lymphocyte count (10^9^/L);Systemic inflammatory index (SII) = neutrophils (10^9^/L) × platelets (10^9^/L)/lymphocytes (10^9^/L);Neutrophil to HDL-C ratio (NHR) = neutrophil count (10^9^/L)/HDL-C (mg/dL);Lymphocyte to HDL-C ratio (LHR) = lymphocyte count (10^9^/L)/HDL-C (mg/dL);Monocytes to HDL-C ratio (MHR) = monocytes count (10^9^/L)/HDL-C (mg/dL);Platelets to HDL-C ratio (PHR) = platelets count (10^9^/L)/HDL-C (mg/dL).

For the cardioembolic risk indexes, we used the formulas stated bellow:TG/HDL-C ratio = total triglycerides (mg/dL)/HDL-C (mg/dL);TC/HDL-C ratio = total cholesterol (mg/dL)/HDL-Cholesterol (mg/dL);non-HDL-C = total cholesterol (mg/dL) − HDL-C (mg/dL);Atherogenic index of plasma = log_10_(TG/HDL-C).

### 2.6. Statistical Analysis

For the initial data gathering, Microsoft^®^ Excel^®^ 2021 MSO (Version 2406 Build 16.0.17726.20078) was used. Statistical analysis of the data were conducted with SPSS 26.0 (SPSS Inc., Chicago, IL, USA).

When in need, English editing was performed using ChatGPT version 4o (OpenAI, San Francisco, CA, USA).

Due to the small size of the group, we used the Shapiro–Wilk test to assess the normality and linearity of the data. This test is a reliable method for determining whether a sample follows a normal distribution, a fundamental assumption for many statistical analyses [49]. Most of the variables did not meet the normality assumptions, likely due to the wide age range of the subjects and the relatively small sample size. Therefore, we employed non-parametric methods to summarize and describe the relationships between the variables.

The following variables were normally distributed: BMI (*p =* 0.052), SBP (*p =* 0.106), DBP (*p =* 0.758), TC (*p =* 0.067), and HDL-C (*p =* 0.181). However, glycemia, triglycerides, all CBC markers except for basophils (*p =* 0.063), and the inflammatory indexes, except for the atherogenic index of plasma (*p =* 0.085) were not normally distributed, with *p*-values less than 0.05.

Continuous variables were represented as mean ± standard deviation (SD), median and interquartile range were expressed as 25th percentile (corresponding to the 1st Quartile) and 75th percentile (corresponding to the 3rd Quartile), while categorical variables were expressed in terms of frequencies or percentages.

All the tables and figures included in the present article were generated and edited using the SPSS software.

## 3. Results

A total of 71 children and adolescents aged between 6 and 16, with a median age of 10.96 ± 2.653 years were included in the study, out of which 67.6% (n = 48) were male and 32.4% (n = 23) were female. All of the included subjects had a waist circumference > the 90th percentile for age and sex. According to the CDC’s classification, 7% (n = 5) of individuals were diagnosed with overweight, 49.3% (n = 35) with obesity, and 43.7% (n = 31) with severe obesity. The medium BMI of all children and adolescents included in the study was 28.15 ± 3.81 kg/m^2^ (Minimum = 21.08 kg/m^2^ and Maximum = 38.982 kg/m^2^). As weight and height are closely dependent on age, we have chosen to use BMI as a more appropriate marker for obesity, being a reliable indicator across different ages. By using BMI, we assess and compare obesity levels more accurately, with no restrictions of the age-related variations of weight and height.

According to the NCEP ATP III criteria adapted for children and adolescents, a positive diagnosis of MetS requires meeting at least three out of five criteria. We found that 45.1% (n = 32) of the subjects in the study met three or more criteria for MetS, placing them in the MetS+ group, while the remaining 54.9% (n = 39) were categorized in the MetS− group. The groups of children and adolescents meeting four or five MetS criteria had the fewest members, with seven (9.9%) and five (7%) individuals, respectively. The best-represented groups, totaling 50.6%, included those who met two criteria (n = 23, 32.4%) and those who met three criteria (n = 20, 28.2%). The group with one MetS criterion included 16 individuals, representing 22.5% of the total. Overall, 45.1% (n = 32) subjects had DBP > 90th percentile, 43.7% (n = 31) had SBP > 90th percentile, 14.1% (n = 10) had basal glucose (glycemia) > 100 mg/dL, and 31% (n = 22) had triglycerides > 110 mg/dL. The most encountered abnormal paraclinical parameter was HDL-C that was found to be less than 40 mg/dL in 45.1% (n = 32) of children and adolescents with overweight or obesity, and, also, in the MetS group it was modified in 25 out of 32 subjects (78.12%). In Appendix A, the mean and median values are presented, along with 25th and 75th percentiles of every parameter according to the presence of the MetS diagnostic criteria.

We determined that gender has no impact on MetS prevalence by using Fisher’s exact test (*p* = 0.802).

We used a Mann–Whitney U test and determined that the MetS+ group had significantly higher values in terms of DBP (*p =* 0.035), SBP (*p =* 0.028), and triglycerides (*p <* 0.001) than those found in the MetS− group, while the HDL-C values (*p <* 0.001) were significantly lower. No significant differences were found in fasting blood glucose (*p =* 0.295) and total cholesterol (*p =* 0.972) between the two groups. For a comprehensive overview of the data, please refer to Appendix A, which provides detailed information and insights, enabling a deeper understanding of the results discussed.

The BMI value in the MetS+ group is a little bit higher but, when it comes to BMI distribution among MetS+ and MetS− groups, there was no statistical difference (*p =* 0.716), as can be observed in Figure 1 and Appendix A.

In Appendix A we presented the mean values of clinical and paraclinical parameters associated with MetS divided by the number of MetS criteria. We can observe there is an unequivocal increase in BMI, DBP, SBP, glycemia, and triglycerides and a decrease in HDL-C values with every increase in the MetS criteria number and, respectively, severity.

A Mann–Whitney U test was performed to evaluate if the distribution of CBC parameters, ESR, inflammatory and cardioembolic indexes in the MetS+ group are different from the ones in the MetS− group. The distribution of leukocytes, lymphocytes, neutrophils, monocytes, platelets, basophiles, NLR, PLR, SII, non-HDL-C, and ESR is the same across categories of MetS, with a *p* > 0.05. On the other hand, the atherogenic index of plasma, TG/HDL-C, TC/HDL-C, MHR, LHR, NHR, and PHR is not evenly distributed among MetS+ and MetS− groups. More details can be found in Appendix A.

Also, children and adolescents with MetS+ had higher values of TG/HDL-C (*p <* 0.001), TC/HDL-C (*p <* 0.001), MHR (*p =* 0.015), LHR (*p =* 0.001), NHR (*p =* 0.001) atherogenic index of plasma (*p <* 0.001), and PHR (*p <* 0.001). The graphical representation can be found in Figure 2.

The Kruskal–Wallis test was conducted to evaluate differences in inflammatory and cardioembolic indexes across groups divided by number of MetS criteria. The findings suggest that the atherogenic index of plasma, TG/HDL-C, TC/HDL-C, MHR, LHR, NHR, and PHR varies significantly across these groups. While, on the other hand, ESR, NLR, PLR, and SII, are not significantly different (*p* > 0.05), when divided according to the number of MetS criteria met.

The pairwise comparison of number of MetS criteria was analyzed and the significance values have been adjusted by the Bonferroni correction for multiple tests. The TG/HDL-C (*p <* 0.001), atherogenic index of plasma (*p <* 0.001), TG/HDL-C (*p* < 0.001), TC/HDL-C (*p <* 0.001), MHR (*p* = 0.003), LHR (*p* = 0.001), NHR (*p* = 0.004), and PHR (*p* < 0.001) were progressively increased with the number of MetS components, as shown in Figure 3.

ROC curve analysis (Figure 4) shows that TG/HDL-C (AUC = 0.877 [95% CI 0.795 to 0.959], *p <* 0.001), the atherogenic index of plasma (AUC = 0.877 [95% CI 0.795 to 0.959], *p <* 0.001), TC/HDL-C (AUC = 0.825 [95% CI 0.726 to 0.924], *p <* 0.001), LHR (AUC = 0.726 [95% CI 0.608 to 0.844], *p =* 0.001), NHR (AUC = 0.731 [95% CI 0.612 to 0.851], *p =* 0.001), and PHR (AUC = 0.776 [95% CI 0.669 to 0.884], *p <* 0.001) had a significant discriminative ability in correctly identify MetS in children and adolescents with obesity, while MHR (AUC = 0.668 [95% CI 0.540 to 0.796], *p =* 0.015) and non-HDL-C (AUC = 0.625 [95% CI 0.492 to 0.758], *p <* 0.071) had a better specificity at higher values.

## 4. Discussion

In this retrospective study, we aimed to evaluate the practicality of using new inflammation and cardioembolic indexes derived from basic paraclinical analysis to distinguish between children and adolescents with different degrees of obesity and MetS. Additionally, we sought to determine the effectiveness of these markers in detecting the low-grade inflammation associated with these conditions. Our goal is to provide clinicians with affordable, easy to reproduce, and available-on-large-scale tools that can be used in everyday clinical settings.

One of the non-communicable diseases that is taking over the world today is obesity [50]. Over the years, various methods were used to assess body weight, but, until today, at populational level, the one that proved to be the most efficient was the BMI [51,52]. In our study, BMI was the tool that helped us divide the subjects into categories based on the severity of obesity, according to BMI percentile, as recommended by the CDC.

Several studies have confirmed that in individuals with obesity, the chronic low-grade inflammation of adipose tissue is linked to metabolic disease and organ tissue complications [53]. Lately, adipose tissue has been considered to be an endocrinologically and metabolically active organ, which is capable of producing cytokines and other inflammatory mediators [54]. These cytokines regulate both the starting and the resolution of inflammation, as well as adaptive and reparative angiogenesis [55]. Weight gain and obesity trigger a phenotypic change characterized by inflamed, dysfunctional adipose cells and the infiltration of immune cells into the stromal vascular fraction [53,56]. Adipocytes release proinflammatory cytokines that disrupt the normal functions of adipose tissue and other organs, so, as a result, it acts like an immune and secretory organ, which is making obesity an inflammatory immune disease [56].

A metanalysis published in 2024 assessing the global prevalence of overweight and obesity across children and adolescents which included 2033 studies from 154 countries reported an 8.5% prevalence across these age groups, with one out of five children being affected by this pathology [57]. Also, they determined that numbers have a great variation when the Human Development index was taken into consideration [57,58]. According to the WHO European Childhood Obesity Surveillance Initiative (COSI), between 2018 and 2020, the prevalence of pediatric overweight was 17.5%, while it was 14.2% for obesity, with boys being the most affected [57,59,60] In our study, which included Caucasian children and adolescents from the southern part of Romania, 7% were diagnosed with overweight, 49.3% with obesity, and 43.7% with severe obesity. Also, when looking at the gender, we can confirm there is a real preponderance of males. This can maybe be explained by the fact that in Romania, children with excess weight are labeled as “fat and beautiful”, especially when it comes to boys, who are not only allowed, but encouraged to be puffier than their female peers, regardless of age.

When it comes to the distribution of gender across the pediatric population with MetS, some studies suggest there is also a greater number of boys with this pathology [61,62], but in the study we conducted, we determined gender had no influence across MetS groups, even if there was a higher number of boys in each category. Research suggests that males may have different genetic predispositions and hormonal responses to obesity, as they tend to accumulate more visceral fat, which is more metabolically active and associated with higher risks of insulin resistance and inflammation [63]. The small sample size and homogeneous population from our study may influence the results we obtained.

Even if the fact that MetS is affecting more and more children and adolescents worldwide, there is not one generally accepted definition for this age group [20], mostly due to the fact that there are controversies regarding the cutoff points [64]. Two of definitions are mostly used in clinical settings: the International Diabetes Federation (IDF) consensus [65] and the National Cholesterol Education Program (NCEP) Adult Treatment Panel-III (ATP-III) criteria modified for children and adolescents [66,67], which we also used in our study as we consider it more fitted for the studied population. The prevalence of MetS in the pediatric population cannot be accurately assessed due to the wide range of diagnostic criteria that can be used [68]. A recent meta-analysis revealed significant differences in the prevalence of MetS among children and adolescents with overweight and obesity that are living in high-income countries, which Romania is also classified as [69]. The prevalence ranged from 25.25% using the IDF criteria to 39.41% using the definition by de Ferranti et al. [70], while the same study gives a 24.47% prevalence when using the NCEP-ATP-III criteria. As expected, due to the fact that we only have included a small number of subjects, which are also diagnosed with obesity, the prevalence of MetS is far greater, with more than half meeting three or more criteria.

As there are five components of MetS, we examined which were the most encountered and determined that HDL-C was by far the most common, not only in the group diagnosed with MetS but also in the non-MetS group. There are some studies in the literature that also spotted this change in cholesterol distribution [66,70]. An explanation for this could be that HDL-C levels are influenced by a poor diet, high in processed foods, and inadequate lifestyle, particularly involving a lack of exercise, which has been shown to be deficient in the pediatric population with weight gain [71,72,73,74].

As expected, and confirmed by other authors too, the levels of blood pressure and triglycerides are higher, and the HDL-C is lower in the MetS group [68,75]. Even though MetS is associated with insulin resistance that in some cases leads to altered blood glucose even to the point of T2DM [76], in our study, there was no statistically significant difference between the two groups. Considering that MetS is defined by five key components, with severity—and therefore cardiovascular risk—likely increasing as more criteria are met [77], we found it valuable to categorize the study subjects according to the number of MetS criteria. The results clearly indicate that as the number of MetS criteria increases, there is a corresponding rise in BMI, DBP, SBP, glycemia, and triglycerides, along with a decrease in HDL-C levels.

Infectious and viral diseases are often underlying causes of chronic illnesses [78,79,80]. Recent data show that systemic low-grade inflammation plays a key role in the development and worsening of chronic diseases [32]. Among the causes of low-grade inflammation are poor dietary habits, particularly those high in processed foods and low in essential nutrients, and lack of physical exercise, that predispose one to obesity [81]. In addition to traditional biomarkers like C-reactive protein [29], CBC derivate indexes are of great value. CBC is the paraclinical investigation that is available everywhere around the world, is cheap and easy to obtain, and the interest of the medical community in its use in detecting cardiovascular risk and inflammation among metabolic diseases, among other things, is obvious [82]. When it comes to other studies on this topic, opinions on CBC parameters and the ratios derived from it, and the ability to detect inflammation or cardiovascular risk in metabolic diseases in adults or children, it is controversial [82,83,84]. When we examined CBC components across MetS groups, as well as within the obesity categories, we found no significant differences in these parameters. Markers such as NLR, PLR, or SII, which were used in other studies on both children and adults and proved to have the capacity to identify the groups at risk [85,86,87], did not perform as well in our study. Regarding the NLR, a recent study determined that this marker is useful in adults but not in children and adolescents [88]. In our opinion, this can be explained by the fact that in the younger population, the inflammatory response is lower than in the adult population. We believe this difference may be due to various factors, including the shorter duration of exposure to metabolic risk factors, differences in immune system function, and the body’s ability to compensate and better adapt in childhood.

On the other hand, some newly introduced inflammatory and cardioembolic indexes used mostly in studies regarding metabolic disorders such as atherogenic index of plasma, TG/HDL-C, TC/HDL-C, MHR, LHR, NHR, and PHR were, in the current research, higher in the group of children and adolescents with MetS, being able to detect a low grade of inflammation and increased risk of developing CVD and T2DM. In a study from a neighboring area of Romania in which children and adolescents with obesity were also evaluated, the researchers highlight the role of these ratios in detecting and monitoring the variation of cardioembolic risk factors [85]. Moreover, regardless of age, these combined inflammatory and cardioembolic indexes proved to be effective in anticipating the dysregulations in the lipid profile and setup of insulin resistance [89,90,91,92].

While low-density lipoprotein cholesterol (LDL-C) is the traditional marker in dyslipidemia management, non-HDL-C has lately been considered a better predictor of cardiovascular disease risk [93,94]. For children with diabetes, the Pediatric NCEP Panel even recommends maintaining non-HDL-C levels below 120 mg/dL [95].

The ROC curve analysis determined the predictive ability and power of TG/HDL-C, the atherogenic index of plasma, TC/HDL-C, LHR, NHR, and PHR when it comes to the presence of MetS. These data are also sustained by other articles recently published with the focus on children and adolescents and also adults [85,96,97]. We determined that LHR rapidly decreases in sensitivity when the thresholds increase, which makes it useful in confirming a MetS positive individual. TC/HDL-C and non-HDL-C are balanced in sensitivity and specificity, a fact that makes them reliable markers for predicting MetS. The recent study by Cura-Esquivel et al. showed that TG/HDL-C and TC/HDL-C significantly increased with the number of MetS criteria. They also found that low-grade inflammation is correlated with MetS in children with excessive weight and that TG, HDL-C, and TG/HDL-C ratio are effective correctly identifying MetS [30].

ESR is an inflammatory marker that is widely used in clinical settings to assess acute conditions as it has a slow increase and remains steady for longer periods of time [98]. Compared to CRP, which is elevated in pediatric patients with MetS [29], ESR is more effective in specific situations like detecting low-grade bone and joint infections and monitoring disease activity in other conditions [98]. Its inability to detect low-grade inflammation in MetS and various stages of obesity was also confirmed in our study, as it showed no correlation with any of the conditions examined.

A limitation of our study is the small sample size. Information about COVID-19 vaccination status was not available in the patients’ files, which is another limitation of our study, as it could influence the immunological parameters. Moreover, the retrospective type is not ideal, as a prospective observational study where the evolution of subjects could be tracked for a longer period of time would add a significant value to the results we obtained. In the future, we plan to increase the number of subjects included. Also, we encourage other researchers to further investigate these inflammatory and cardioembolic indexes, which have proven to be valuable in assessing inflammation and future cardiovascular risk in both MetS and obesity in children and adolescents. It would be interesting to explore the relationship between these markers and dietary patterns and also have a control group of children and adolescents that are healthy.

Ultimately, we wish to emphasize the utility of using inflammatory and cardioembolic indices derived from basic paraclinical tests in assessing patients with obesity and MetS. We also encourage future researchers to explore the utility of these markers in monitoring the progress of patients undergoing both pharmacological and non-pharmacological interventions. These are great tools for clinicians due to their reproducibility, easy access, low-cost, and the possibility to use them regardless of age and sex.

## 5. Conclusions

In the current study, we indicated that the novel cardioembolic and inflammatory indexes, such as MHR, NHR, LHR, NHR, PHR, TG/HDL-C, the atherogenic index of plasma, and TC/HDL-C, are useful in assessing obesity and metabolic syndrome in children and adolescents. These indexes are particularly valuable due to the fact that they are easy to reproduce and use in clinical settings, widely available, and cost-effective, which makes them ideal for use in clinical settings.

Nonetheless, we found that widely used inflammatory indexes such as NLR, PLR, SII, and ESR—which have shown promising results in adults and in some studies involving the pediatric population—were not useful in differentiating MetS or varying degrees of obesity severity in our studied group.

However, our study is preliminary, and larger-scale studies are needed to further assess these indexes. Given the relatively small size of our cohort, the results are not generalizable to the broader population.

## Figures and Tables

**Figure 1 life-14-01206-f001:**
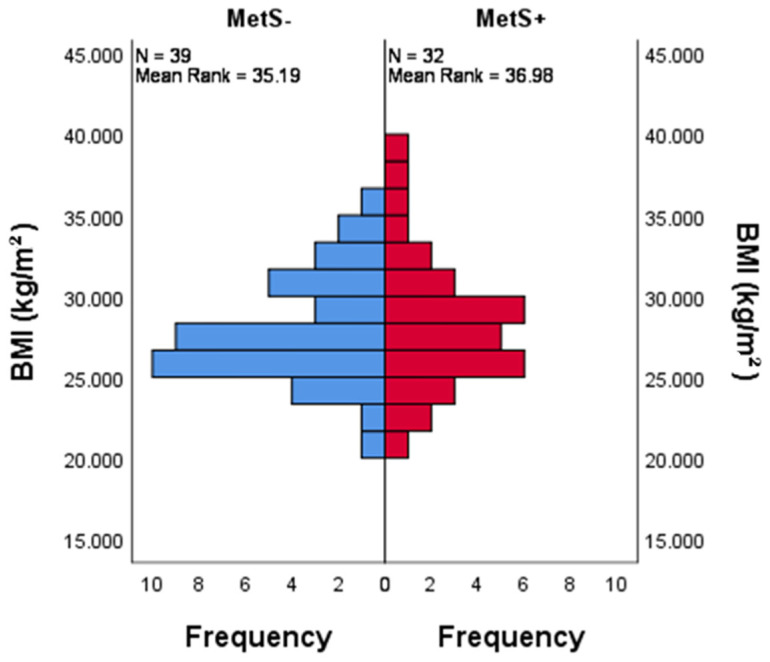
The distribution of BMI across categories of MetS. Bar chart showing there is an even distribution of BMI among the MetS+ and MetS− groups, which was determined using the non-parametric Mann–Whitney U test. *p =* 0.716.

**Figure 2 life-14-01206-f002:**
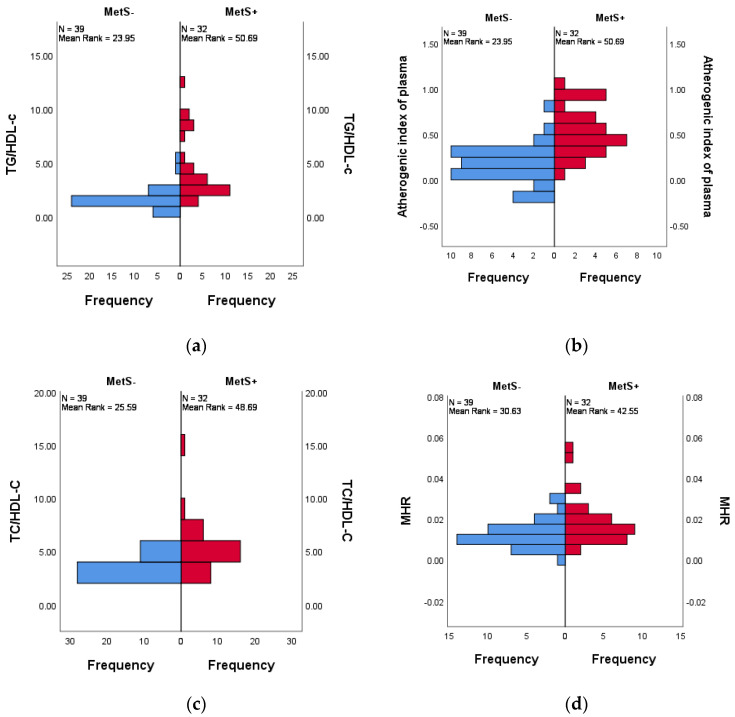
Distribution across MetS+ and MetS− groups of (**a**) TG/HDL-C (*p <* 0.001), (**b**) atherogenic inflammatory indexes (*p <* 0.001), (**c**) TC/HDL-C (*p <* 0.001), (**d**) MHR (*p =* 0.015.), (**e**) LHR (*p =* 0.001), (**f**) NHR (*p =* 0.001), (**g**) PHR (*p <* 0.001). The difference between the MetS groups for each index was determined using the non-parametric Mann–Whitney U test, with a statistical significance at *p <* 0.05.

**Figure 3 life-14-01206-f003:**
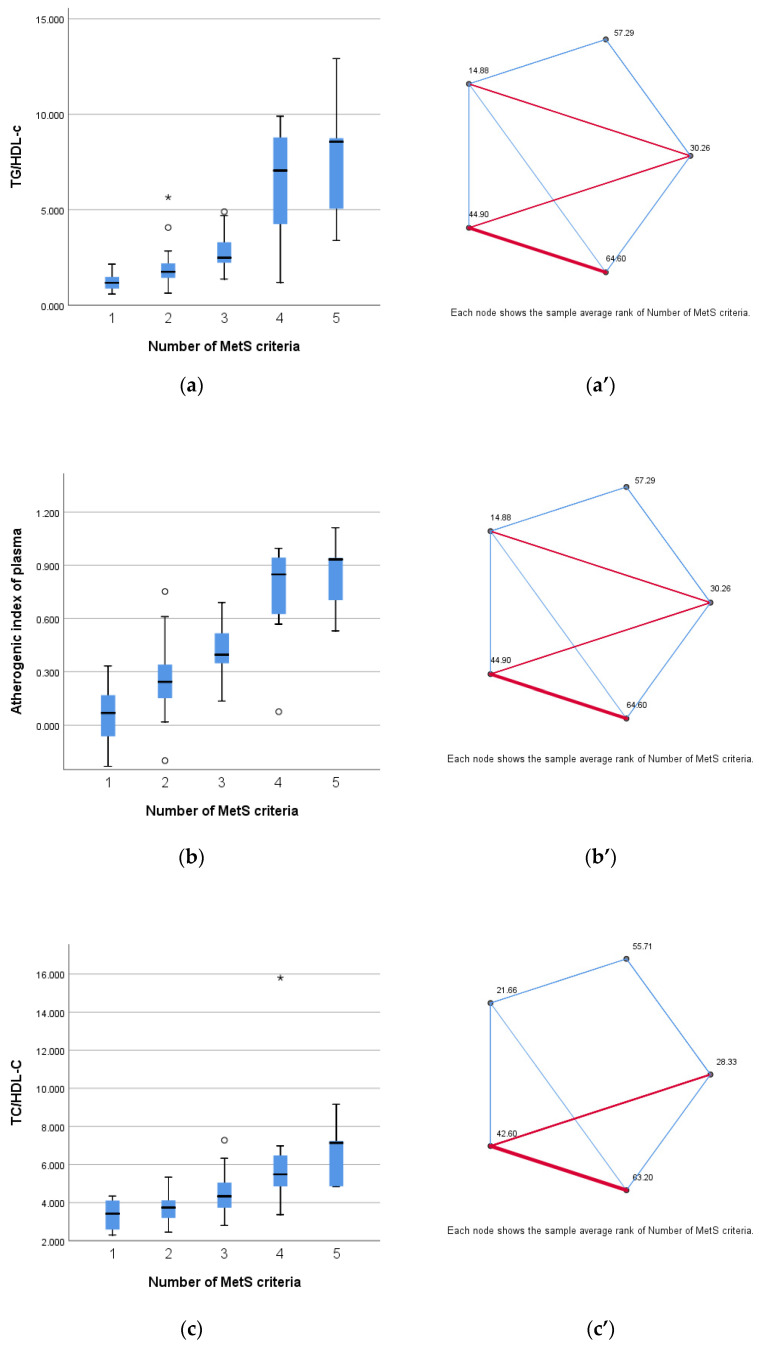
Distribution according to number of MetS number of criteria of (**a**) TG/HDL-C (*p* < 0.001), (**b**) atherogenic inflammatory index (*p* < 0.001), (**c**) TC/HDL-C (*p* < 0.001), (**d**) MHR (*p* = 0.003), (**e**) LHR (*p* = 0.001), (**f**) NHR (*p* = 0.004), and (**g**) PHR (*p* < 0.001). (**a’**–**g’**) Pairwise comparison of number of MetS criteria for each parameter. The nodes connected by red lines suggest a stronger association between the parameters assessed and the categories divided according to number of MetS criteria, while the blue lines indicate a weaker association between the before mentioned groups. The difference between the MetS groups for each index was determined using the non-parametric Mann–Whitney U test, with a statistical significance at *p* < 0.05.

**Figure 4 life-14-01206-f004:**
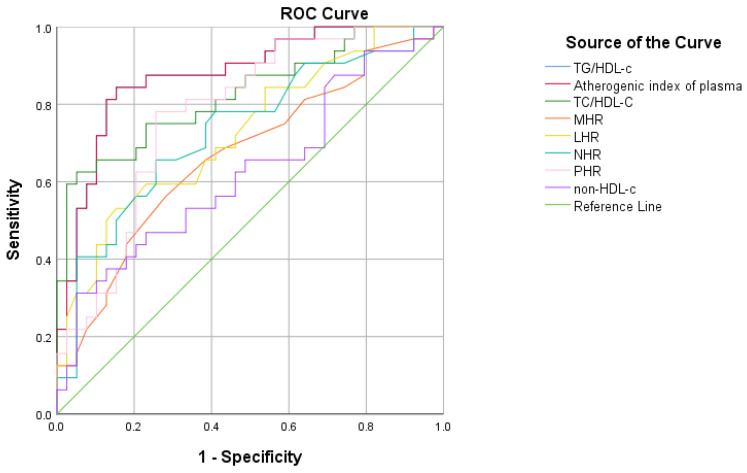
ROC curve analysis to evaluate the diagnostic performances of inflammatory and cardioembolic indexes in MetS in children and adolescents with obesity.

## Data Availability

The original contributions presented in the study are included in the article; further inquiries can be directed to the corresponding author.

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
