# Peer review of "The Role of Paraclinical Investigations in Detecting Inflammation in Children and Adolescents with Obesity and Metabolic Syndrome"

_life, 2024, doi:10.3390/life14091206_

Round 1

Reviewer 1 Report

Comments and Suggestions for Authors

In this study (Life-3204903), Podeanu et al., performed retrospective analysis to evaluate the effectiveness of several routinely used markers to detect obesity and metabolic syndrome in young populations. Given the rising global epidemiological trend of obesity and metabolic diseases such studies hold significant clinical value. However, the present manuscript raises several concerns.

 /span/p p class="Default" style="margin-left: .25in; text-align: justify"span Concerns:/span/p p class="Default" style="margin-left: .5in; text-align: justify; text-indent: -.25in; mso-list: l0 level1 lfo1"1.      The entire text, starting from the abstract, needs to be very carefully edited for the language and spelling/orthography. The use of the AI (Chat GPT; as mentioned by the authors) did not seem to be very helpful.

2.      The patient cohort utilized in this study was from January 2021 to July 2024. During most of the 2021 the World was experiencing SARS COV-19. Does any of the included members infected and/or vaccinated? This will have a significant influence on immunological parameters.

3.      The authors should perform additional statistical analyses by segregating the groups based on the status of puberty. The text should clearly mention the numbers of patients who reached puberty.

4.      Was HOMA-IR performed? Insulin levels? The study was done after 8 hours of fasting-is its post-dinner or any other time? When during the day the blood was collected?

5.      Information on LDL levels is also needed.

6.      Authors noted a significant correlation between obesity and gender, which was not so for MetS. Can authors speculate more about the molecular causes for this along with the possibility of a very small and ethnically homogenous population used in this study.

Comments on the Quality of English Language

See my comments to the authors.

Author Response

Response to Reviewer 1 X Comments

1. Summary

Thank you for taking the time to review our manuscript. We greatly appreciate your valuable input in evaluating our retrospective analysis. As the global prevalence of obesity and metabolic syndrome continues to rise, we believe studies like ours hold significant clinical value. We have carefully considered your concerns and have made the necessary corrections. Detailed responses to your feedback are provided below, along with the corresponding changes made using the track changes option in the resubmitted files. Your insights have been instrumental in improving the quality of our manuscript, and we are grateful for your thorough work.

2. Questions for General Evaluation

Reviewer’s Evaluation

Response and Revisions

Does the introduction provide sufficient background and include all relevant references?

Yes

We have made several revisions throughout the manuscript. Please refer to the final uploaded version to review the changes in detail.

Are all the cited references relevant to the research?

Yes

Is the research design appropriate?

Can be improved

Are the methods adequately described?

Can be improved

Are the results clearly presented?

Can be improved

Are the conclusions supported by the results?

Can be improved

3. Point-by-point response to Comments and Suggestions for Authors

Comment 1: The entire text, starting from the abstract, needs to be very carefully edited for the language and spelling/orthography. The use of the AI (Chat GPT; as mentioned by the authors) did not seem to be very helpful.

Response 1: Thank you for your feedback. We have made several adjustments to the text in order to improve language and spelling throughout the manuscript. Additionally, we sought external validation to ensure the final version meets the required standards. All changes have been made using track changes option. Since the revisions are quite extensive, we won't list each of them here, but they can be check in the manuscript as stated above. We appreciate your thorough review and are committed to enhancing the quality of our work.

Regarding the title of the article, see fitted to change it to: “The Role of Paraclinical Investigations in Detecting Inflammation in Children and Adolescents with Obesity and Metabolic Syndrome”.

Original title: “The Role of Paraclinical Investigations in Uncovering Inflammation in Children and Adolescents with Obesity, With or Without Metabolic Syndrome”.

Comment 2: The patient cohort utilized in this study was from January 2021 to July 2024. During most of the 2021 the World was experiencing SARS COV-19. Does any of the included members infected and/or vaccinated? This will have a significant influence on immunological parameters.

Response 2:  Thank you for your insightful comment. None of the patients included in our study had an acute infection. We also stated it in the chapter 2.2. Subjects included in the study (“None of the children and adolescents included in the study were under any medication or diagnose with any other acute or chronic condition at the moment of the examination according to the data stated in the patient`s file.”). However, we acknowledge that we did not identify in the patient’s medical record data regarding COVID-19 vaccination status.

We made some changes in the manuscript to address the issues stated above, as it follows:

Chapter: 2.2. Subjects included in the study

Original:” None of the children and adolescents included in the study were under any medication or diagnose with any other acute or chronic condition at the moment of the examination according to the data stated in the patient`s file.”

Changed: ”None of the children and adolescents included in the study were under any medication or diagnose with any other acute or chronic condition, including COVID-19, at the moment of the examination according to the data stated in the patient`s file.”

Chapter Discussion :

Added: ” Information about COVID-19 vaccination status was not available in the patients' files, which is another limitation of our study, as it could influence the immunological parameters.”

Comment 3: The authors should perform additional statistical analyses by segregating the groups based on the status of puberty. The text should clearly mention the numbers of patients who reached puberty.

Response: Thank you for your suggestion. This is a very interesting perspective. However, performing additional statistical analyses based on puberty status would extend the scope of the study significantly. Early puberty is commonly observed in obese patients, and there are many existing studies that does not take it into consideration. If you believe it is crucial to include data on puberty status, we will consider adding this information and comment the results comparing them to exiting data in the field. Please, let us know your opinion on this matter as soon as possible.

Comment 4:   Was HOMA-IR performed? Insulin levels? The study was done after 8 hours of fasting-is its post-dinner or any other time? When during the day the blood was collected?

Response 4: You made a very good point. HOMA-IR and insulin levels were not included in the study as the glycemic values did not require it. Additionally, it is important to consider the retrospective nature of the study, which limited our ability to influence the availability of laboratory analyses. Blood samples were collected after an at least 8-hour fasting period, in the morning. We also, added this information in the text.

Original:

“A blood sample was obtained after a minimum of 8 hours of fast by phlebotomy by a qualified nurse.”

Changed:

“A blood sample was obtained after a minimum of 8 hours of fast, in the morning, by phlebotomy by a qualified nurse.”

Comment 5: Information on LDL levels is also needed.

Response 5: Considering that criteria for metabolic syndrome do not include LDL-C, we did not find it necessary to be included in the study. However, we will take into account your suggestion and add it in future studies.

Comment 6:  Authors noted a significant correlation between obesity and gender, which was not so for MetS. Can authors speculate more about the molecular causes for this along with the possibility of a very small and ethnically homogenous population used in this study.

Response 6: Thank you for addressing this topic. We want to point out the fact that we did not used the term “correlation” when it comes to either obesity and gender, or metabolic syndrome and gender, but prevalence and distribution (Chapter Discussion: “Also, when looking at the gender, we can confirm there is a real preponderance of males.” and “When it comes to the distribution of gender across the pediatric population with MetS, some studies suggest there is also a greater number of boys affected by this pathology [62,63], but in the study we conducted, we determined gender had no influence across MetS groups, even if there was a higher number of boys in each category.”).

Regarding the population and sample size, as you pointed, we specified in the text that also the homogeneity of the population may play a role:

Original:

“The small sample size could be a big factor for these results.”

Changed:

“The small sample size and homogeneous population from our study may influence the results we obtained.”

Also, we added some comments regarding the cause for this gender inequality.

Added: “Research suggests that males may have different genetic predispositions and hormonal responses to obesity, as they tend to accumulate more visceral fat, which is more metabolically active and associated with higher risks of insulin resistance and inflammation [64].”

Reference:

64. Koceva, A.; Herman, R.; Janez, A.; Rakusa, M.; Jensterle, M. Sex- and Gender-Related Differences in Obesity: From Pathophysiological Mechanisms to Clinical Implications. Int J Mol Sci 2024, 25, doi:10.3390/ijms25137342.

We hope that with these changes, we have successfully addressed your helpful suggestions.

4. Response to Comments on the Quality of English Language

Point 1: Extensive editing of English language required.

Response 1: Thank you for your feedback. We made some changes to improve the language quality as suggested. Your input is appreciated.

Reviewer 2 Report

Comments and Suggestions for Authors

I congratulate the Authors for their original research.

I have some advices that I hope can improve the quality of the paper 

minor comments:

Line 18 please rephrase the first sentence, including that the prevalence of MS is increasing

line 22 retrospective what? I think you miss some words

line 385 please specify that BMI works very well but only at population level

line 407 please rephrase, the sentence is not clear about prevalences

line 439 what is "reach"?

line 454 Please add a mention to poor diet as cause of low grade inflammation and chronic diseases

line 467 please remove the comma

line 469 please change the term "believe" to don't repeat

line 472 Is there a reference to your idea or is only your opinion?

line 525 I think "reproductible" is not correct

other suggestions:

Please don't be so much enthusiastic in your conclusions, the study is only a preliminary findings that, as you said, must be confirmed in other bigger studies

Comments on the Quality of English Language

The language can be improved as I suggested in my report, some minor corrections are required

Author Response

Response to Reviewer 2 X Comments

1. Summary

Thank you very much for taking the time to review our manuscript. We appreciate your valuable input in helping us elevate the quality of our work. We have carefully considered your suggestions and made the necessary corrections. Please find detailed responses to your feedback below, along with the corresponding changes marked using the track changes option in the resubmitted manuscript. Your insights have been instrumental in improving our work and we are grateful for your contribution.

2. Questions for General Evaluation

Reviewer’s Evaluation

Response and Revisions

Does the introduction provide sufficient background and include all relevant references?

Yes

We have made several revisions throughout the manuscript. Please refer to the final uploaded version to review the changes in detail.

Are all the cited references relevant to the research?

Yes

Is the research design appropriate?

Yes

Are the methods adequately described?

Yes

Are the results clearly presented?

Yes

Are the conclusions supported by the results?

Can be improved

3. Point-by-point response to Comments and Suggestions for Authors

Comments:

1.     Line 18 please rephrase the first sentence, including that the prevalence of MS is increasing

2.     line 22 retrospective what? I think you miss some words

3.     line 385 please specify that BMI works very well but only at population level

4.     line 407 please rephrase, the sentence is not clear about prevalences

5.     line 439 what is "reach"?

6.     line 467 please remove the comma

7.     line 469 please change the term "believe" to don't repeat

Response: Agree. In the revised manuscript, we have thoroughly addressed and incorporated all the suggested changes provided. You can check the final manuscript to ensure they were made appropriately. We sincerely appreciate your meticulous attention to detail and the effort you’ve invested in helping us refine our work. Your feedback has been invaluable in ensuring the accuracy and quality of our research.

Comment: line 454: Please add a mention to poor diet as cause of low grade inflammation and chronic diseases

Response: Thank you for the suggestion. We have now added a mention addressing that a poor diet can contribute to low-grade inflammation and the development of chronic diseases, by changing the next paragraph:

Original: “Infectious and viral diseases are often underlying causes of chronic illnesses [79–81], but recent data show systemic inflammation plays a key role in the development and worsening of chronic diseases [32].”

Changed: “Infectious and viral diseases are often underlying causes of chronic illnesses [79–81].  Recent data show systemic low-grade inflammation plays a key role in the development and worsening of chronic diseases [32]. Among the causes of low-grade inflammation are poor dietary habits, particularly those high in processed foods and low in essential nutrients, and lack of physical exercises, that predispose to obesity [82].

Reference:

82. Hart, M.J.; Torres, S.J.; McNaughton, S.A.; Milte, C.M. Dietary Patterns and Associations with Biomarkers of Inflammation in Adults: A Systematic Review of Observational Studies. Nutr J 2021, 20, 24, doi:10.1186/s12937-021-00674-9.

We appreciate your input in helping us improve the manuscript.

Comment: line 472 Is there a reference to your idea or is only your opinion?

Response: It reflects our opinion on the matter. At the moment, we do not have a specific reference to support this idea, as we have not found convincing evidence on this subject. Our view is based on our current understanding and analysis, but we acknowledge that additional research may be needed to provide a more solid statement.

Comment: line 525 I think "reproductible" is not correct.

Response: Agree. We have revised that particular issue and we made some changes.

Original: “These indexes are particularly valuable due to the fact that are reproductible, easy to use in clinical settings, have a great availability all over the world, and determining all of them costs a fair amount of money, making them ideal to be used in clinical settings.”

Changed: “These indexes are particularly valuable due to the fact that are easy to reproduce and use in clinical settings, widely available and cost-effective, which makes them ideal to be used in clinical settings.“

Comment: Please don't be so much enthusiastic in your conclusions, the study is only a preliminary findings that, as you said, must be confirmed in other bigger studies

Response: Thank you for your feedback. We appreciate the reminder to temper our conclusions. We agree that our findings are preliminary and should be validated through larger studies before drawing definitive conclusions, as we also mentioned in the last paragraph. Moreover, we revised our statements to better reflect the preliminary nature of the study.

Original:

“In the current study, we determined that the novel cardioembolic and inflammatory indexes such as MHR, NHR, LHR, NHR, PHR, TG/HDL-C, atherogenic index of plasma, TC/HDL-C are valuable in assessing obesity and metabolic syndrome in chil-dren and adolescents. These indexes are particularly valuable due to the fact that are reproductible, easy to use in clinical settings, have a great availability all over the world, and determining all of them costs a fair amount of money, making them ideal to be used in clinical settings.

Nonetheless, we found that widely used inflammatory indexes such as NLR, PLR, SII, and ESR— which have shown promising results in adults and some studies in-volving the pediatric population— were not useful in differentiating MetS or varying degrees of obesity severity in our studied group.

However, more studies should focus on assessing these indexes, as our cohort was relatively small, and therefore, the results may not be generalizable to the broader population.”

Changed:

“In the current study, we indicated that the novel cardioembolic and inflammatory indexes such as MHR, NHR, LHR, NHR, PHR, TG/HDL-C, atherogenic index of plasma, TC/HDL-C are useful in assessing obesity and metabolic syndrome in children and adolescents. These indexes are particularly valuable due to the fact that are easy to reproduce and use in clinical settings, are widely available and cost-effective, which makes them ideal to be used in clinical settings.

Nonetheless, we found that widely used inflammatory indexes such as NLR, PLR, SII, and ESR— which have shown promising results in adults and some studies involving the pediatric population— were not useful in differentiating MetS or varying degrees of obesity severity in our studied group.

However, our study is preliminary, and larger-scale studies are needed to further assess these indexes. Given the relatively small size of our cohort, the results are not generalizable to the broader population.”

4. Response to Comments on the Quality of English Language

Point 1: The language can be improved as I suggested in my report, some minor corrections are required

Response 1: Thank you for your feedback. We made some changes to improve the language quality as suggested. Your input is appreciated.

Reviewer 3 Report

Comments and Suggestions for Authors

A patient with metabolic syndrome is a high-risk cardiovascular patient. The authors examine whether existing paraclinical indicators reflect inflammation present in the metabolic syndrome or obesity itself. So it seems that the idea is not entirely new.

It raises a doubt in my mind : in line 105 it was described that the search criterion was BMI > 87 th percentile; why if the obesity criterion> 85 th perc- line 158.

line 183 is wrong blood glucose value - should be 100 mg/dl not 110 mg/dl

Referring to and explaining statistical methods when presenting results causes confusion. It would be better to separate these parts definitively.

Tables should be placed in Supplementary Materials.

The discussion is too long , and not focused enough on convincing the reader of the need to determine the indicators presented.

 Sentence 517. indicators to monitor Mes or maybe rather to monitor non-pharmacological interventions ? How will the studied indicators work in patients who will require pharmacological intervention ? please address this in the conclusions.

Author Response

Response to Reviewer 3 X Comments

1. Summary

Thank you very much for taking the time to review our manuscript. We truly appreciate your valuable input in helping us enhance the quality of our work. We have carefully reviewed your suggestions and implemented the necessary revisions. Detailed responses to your feedback are provided below, and the corresponding changes have been marked using the track changes option in the resubmitted manuscript. Your insights have been instrumental in improving our study, and we are sincerely grateful for your contribution.

2. Questions for General Evaluation

Reviewer’s Evaluation

Response and Revisions

Does the introduction provide sufficient background and include all relevant references?

Must be improved

We have made several revisions throughout the manuscript. Please refer to the final uploaded version to review the changes in detail.

Are all the cited references relevant to the research?

Must be improved

Is the research design appropriate?

Can be improved

Are the methods adequately described?

Must be improved

Are the results clearly presented?

Must be improved

Are the conclusions supported by the results?

Can be improved

3. Point-by-point response to Comments and Suggestions for Authors

Comment 1: A patient with metabolic syndrome is a high-risk cardiovascular patient. The authors examine whether existing paraclinical indicators reflect inflammation present in the metabolic syndrome or obesity itself. So, it seems that the idea is not entirely new.

Response 1: Indeed, we acknowledge that the concept of using paraclinical indicators to reflect inflammation in metabolic syndrome or obesity is not entirely new. However, the focus of our study was to determine whether these markers are reliable in differentiating between varying degrees of obesity and metabolic syndrome in children and adolescents. We might add, that even if these markers have been explored in adult populations, their use in pediatric populations is relatively new and requires further validation in studies to establish their reliability and effectiveness in different age groups and situations.

Comment 2: It raises a doubt in my mind : in line 105 it was described that the search criterion was BMI > 87 th percentile; why if the obesity criterion> 85 th perc- line 158.

Response 2: Thank you for pointing that out. We acknowledge that this was a mistake in our manuscript. The correct criterion should consistently refer to BMI > 85th percentile for defining obesity, as described in line 158. We will correct the discrepancy in line 105 to ensure the criteria are accurate and consistent throughout the manuscript. We are extremely grateful for your attention to details.

Comment 3: line 183 is wrong blood glucose value - should be 100 mg/dl not 110 mg/dl.

Response 3: Thank you again for your careful attention to this detail. We have corrected the value. Although we mistakenly wrote 110 mg/dL, we used the correct value of 100 mg/dL in processing the results. The manuscript has been updated to reflect this correction.

Comment 4: Referring to and explaining statistical methods when presenting results causes confusion. It would be better to separate these parts definitively.

Response 4: Thank you for your suggestion. In our approach, we followed the structure commonly used in similar articles within the field, where statistical methods are referenced alongside the results for clarity. We believe this helps the reader better understand how the data were analyzed and interpreted. However, we made some changes in the results section as you suggested.

Comment 5: Tables should be placed in Supplementary Materials.

Response 5: Thank you for your suggestion. We will move the tables to the Supplementary Materials as requested.

Comment 6: The discussion is too long , and not focused enough on convincing the reader of the need to determine the indicators presented.

Response 6: Thank you for your feedback. We agree that the discussion could benefit from being more focused and concise. To address this, we have made several revisions to better emphasize the importance of determining the indicators presented. As the changes are quite extensive, we have incorporated them directly into the manuscript. You will be able to review all the adjustments in the revised version of the manuscript.

Comment 7: Sentence 517. indicators to monitor Mes or maybe rather to monitor non-pharmacological interventions ? How will the studied indicators work in patients who will require pharmacological intervention ? please address this in the conclusions.

Response 7: Thank you for your very interesting perspective. Evaluating whether the studied indicators are also useful in monitoring pharmacological interventions is indeed an important question. However, it is beyond the scope of our current study. Reviewing the mentioned statement, we can understand it causes a bit of confusion. We intended it as more of an idea for future studies. We changed it accordingly as it follows: 

Original: “Ultimately, we wish to emphasize the utility of using inflammatory and cardioembolic indices derivate from basic paraclinical tests in assessing and, maybe, monitoring the evolution of patients with obesity and MetS.”

Changed: “Ultimately, we wish to emphasize the utility of using inflammatory and cardioembolic indices derivate from basic paraclinical tests in assessing patients with obesity and MetS. We also encourage future researchers to explore the utility of these markers in monitoring the progress of patients undergoing both pharmacological and non-pharmacological interventions. “

Round 2

Reviewer 3 Report

Comments and Suggestions for Authors

  I accept

Author Response

1. Questions for General Evaluation

Reviewer’s Evaluation

Response and Revisions

Does the introduction provide sufficient background and include all relevant references?

Yes

Thank you!

We made some changes in the attached manuscript.

Are all the cited references relevant to the research?

Yes

Is the research design appropriate?

Yes

Are the methods adequately described?

Yes

Are the results clearly presented?

Can be improved

Are the conclusions supported by the results?

Yes

2. Point-by-point response to Comments and Suggestions for Authors

Comments 1: I accept.

Response 1: We sincerely thank you for the thorough and detailed work you have put into reviewing our manuscript. Your valuable feedback and suggestions have been instrumental in helping us improve the overall quality and clarity of the paper. We truly appreciate your effort and time in assisting us throughout this process. 
